# Prevalence, Risk Factors and Course of Osteoporosis in Patients with Crohn’s Disease at a Tertiary Referral Center

**DOI:** 10.3390/jcm8122178

**Published:** 2019-12-10

**Authors:** Peter Hoffmann, Johannes Krisam, Christian Kasperk, Annika Gauss

**Affiliations:** 1Department of Gastroenterology and Hepatology, University Hospital Heidelberg, INF 410, 69120 Heidelberg, Germany; annika.gauss@med.uni-heidelberg.de; 2Department of Medical Biometry, Institute of Medical Biometry and Informatics, University Hospital Heidelberg, INF 130.3, 69120 Heidelberg, Germany; krisam@imbi.uni-heidelberg.de; 3Department of Endocrinology and Clinical Chemistry, University Hospital Heidelberg, INF 410, 69120 Heidelberg, Germany; Christian.Kasperk@med.uni-heidelberg.de

**Keywords:** bone mineral density, osteopenia, osteoporosis, Crohn’s disease, inflammatory bowel disease, steroids, bisphosphonates

## Abstract

Background: Patients with Crohn’s disease are at increased risk for fractures due to low bone mineral density (BMD). Real-world data are necessary to optimize surveillance and treatment strategies. Methods: Patients with Crohn’s disease who underwent at least one dual-energy X-ray absorptiometry (DXA) scans were recruited. The primary study endpoints were (1) prevalence of osteoporosis, and (2) factors influencing changes of BMD. To identify potential risk factors for reduced BMD, Mann–Whitney U-test was used for ordinal and continuous variables and *x*²-tests for categorical variables. Results with *p* < 0.05 were included in a multivariable analysis. To identify potential factors influencing changes in BMD, a generalized linear mixed model was applied. Results: 39.9% of the patients were diagnosed with normal BMD, 40.2% with osteopenia, and 19.8% with osteoporosis. The main risk factors for osteoporosis were low body mass index (BMI), previous bowel resections and male sex. The main risk factors for reduced BMD during further along the disease course were steroid use, history of immunomodulator treatment, female sex and decreased BMI. Conclusion: Low BMI, previous bowel resections and male sex were the main risk factors for the development of osteoporosis. Steroid use reduced BMD even under anti-inflammatory therapy, underlining that they should be used with great care in that patient group.

## 1. Introduction

Inflammatory bowel diseases (IBDs)—encompassing Crohn’s disease (CD) and ulcerative colitis (UC)—are known to be associated with decreased bone mineral density (BMD) [1,2]. Even though there has been some controversy about whether the prevalence of bone fractures is also increased in that population, a recent meta-analysis revealed that the risk for bone fractures in IBD patients was 38% higher than in the general population [3]. Decreased BMD is diagnosed in up to 30% of IBD patients [4]. The mean BMD of IBD patients was 10% lower than that of the general population [5]. The prevalence of osteoporosis in the IBD population ranges from 18% to 42%, whereas the prevalence of osteopenia ranges from 22% to 77% [6]. IBD patients, in particular, have been shown to carry an increased risk for vertebral fractures [7,8]. It has also been demonstrated that low BMD frequently occurs in patients after restorative proctocolectomy [9].

The pathogenesis of the loss of BMD in IBD patients is multifactorial. Among previously identified risk factors are activity and severity of IBD, fistulizing disease course in CD, low food intake and malabsorption with vitamin D deficiency, as well as low body mass index (BMI), intestinal surgery, corticosteroid treatment and smoking [10,11]. Also, male IBD patients showed a trend toward lower BMD [12]. Furthermore, genetic factors are discussed [13]. For example, there are suggestions that interleukin 6 might negatively regulate osteoblast differentiation, while NOD2 mutations did not confer a risk for loss of BMD [12,14]. The use of corticosteroids induces the apoptosis of osteoblasts [15]. However, low BMD is already present in newly diagnosed CD patients without previous corticosteroid therapy [14], so CD itself seems to negatively affect the bone mass [16].

It has been demonstrated that only 22% of newly diagnosed IBD patients have normal serum vitamin D levels [17]. Another study identified low serum vitamin D levels, male sex, diagnosis of CD, corticosteroid therapy and Asian origin as risk factors for low BMD in IBD patients [18]. The duration of IBD is an independent predictor of low BMD [19].

Low BMD, in general, implies a higher risk of fractures. A BMD value of *T*-score < −1 implies an increased relative risk of 2.6 [20,21].

It has been proven in high-risk patients such as postmenopausal women, or patients treated with long-term corticosteroids, that an adequate treatment of low BMD may prevent osteoporotic fractures, so it is of great importance to identify patients at risk of low BMD early in the disease course and to promptly initiate a suitable therapy [22,23,24]. This is even more essential in the young population of IBD patients. So far, few studies have dealt with the impact of novel IBD therapies on the individual courses of BMD. One of the reasons for this lack is that BMD is infrequently monitored on a regular basis in IBD patients, especially when they are very young.

Krajcovicova et al. have recently demonstrated that combination therapy with an immunomodulator and an anti-TNFα (tumor necrosis factor α) agent improved BMD in IBD patients. This is another rationale for early aggressive anti-inflammatory treatment of IBD patients with high disease activity and the risk of a complicated disease course [19].

Based on these observations, the main goals of the present study were to explore the prevalence and risk factors of low BMD in patients with CD, and to describe BMD courses during different modern IBD therapies. The superordinate aim of this project is to improve the prevention and therapy of low BMD in the identified population at risk, and thus to reduce the burden of osteoporotic fractures in CD patients.

## 2. Materials and Methods

### 2.1. Study Design and Data Extraction

This is an uncontrolled, single-center retrospective study including outpatients with a diagnosis of CD at a German university hospital serving as a tertiary referral center for the treatment of IBD. The study was approved by the Ethics Committee of the University of Heidelberg (protocol number: S-354/2017). Inclusion criteria were as follows: (1) age ≥ 18 years, (2) diagnosis of CD according to ECCO criteria [25], (3) being a patient at the IBD outpatient clinic of the University Hospital Heidelberg, (4) at least one visit to the IBD outpatient clinic between 1 June 2014 and 30 June 2017 and (5) at least one documented dual-energy X-ray absorptiometry (DXA) performed prior to 30 June 2017, as this date was defined as the cut-off time point for data acquisition.

All data were retrieved from entirely computerized medical records. To identify eligible individuals, all electronically available daily appointment lists of the IBD outpatient clinic between 1 June 2014 and 30 June 2017 were screened.

Bone masses, *T*-values and BMI, as well as their changes compared to prior BMD results (indicated in percent per year), were extracted directly from the BMD result sheets. Age at diagnosis of CD and at first BMD measurement, sex, smoking habits, extraintestinal manifestations of CD, disease duration until first BMD measurement, CD location and phenotype, intestinal operations, presence of ostomy, short bowel syndrome, nutrition, medications, treatment changes and reasons for treatment changes as well as steroid therapy and the therapy for osteopenia and osteoporosis were recorded.

Demographic and disease-related parameters of all eligible patients were entered into a Microsoft Excel spreadsheet. At the outpatient clinic, all IBD patients who presented more than once underwent a routine BMD measurement shortly after their first presentation. Patients who visited the clinic only for a second opinion did not undergo DXA. In patients with diagnosed osteopenia, further BMD measurements were performed every two years. Patients with diagnosed osteoporosis underwent further BMD measurements every year. As a standard of care, patients received calcium and vitamin D whenever they were on corticosteroid therapy, or when either osteopenia or osteoporosis was diagnosed by BMD measurement. In addition, a bisphosphonate-based therapy was individually started in patients with diagnosed osteoporosis, depending on their *T*-values and further risk factors such as ongoing steroid therapy.

### 2.2. Definitions

The Montreal classification for CD [26] was applied to categorize the disease extent. A steroid-treated patient was defined as a patient for whom at least one treatment phase with oral or intravenous systemic corticosteroids was documented between two DXA scans. Patients receiving budesonide therapy were not counted as steroid-treated patients due to the high first-pass effect of budesonide. The precise duration of the treatment phase was not defined in this study due to its retrospective character, so the steroid treatment duration could not be included in the analyses.

Osteopenia is defined as *T*-values between −2.5 and −1.0, while osteoporosis is defined as *T*-values < −2.5, meaning that BMD differs < 2.5 standard deviations (SD) from the BMD of a woman aged 20–29 years [27]. For patients aged < 20 years, where no *T*-values are validated, a *Z*-value < −1.0 was applied to define osteopenia, and a *Z*-value < −2.5 to define osteoporosis. On a routine basis, both the BMD of the lumbar spine and the left femur were measured, but only the worse result per case was included in the analyses.

### 2.3. Definitions of Variables

Studied variables included age, sex, disease extent, age at first diagnosis of CD, stenosis or fistula, abdominal operations and history of abdominal operations, ostomy, short bowel syndrome, corticosteroid use, BMI and changes in BMI, smoking, IBD therapy and history of IBD therapy, therapy of osteopenia and osteoporosis and history of osteopenia and osteoporosis therapy (calcium and vitamin D, bisphosphonate, antibody or teriparatide therapy), bone mass, T- and *Z*-value and changes in bone mass over time.

Change of BMD is defined as the average of changes in BMD of the lumbar spine and femoral measurements.

### 2.4. Dual X-Ray Energy Absorptiometry (DXA)

All included patients underwent BMD measurement by dual X-ray energy absorptiometry (DXA) (Discovery W, QDR, series, Hologic Waltham, MA, USA) in the lumbar spine (L1–L4) and femoral neck. BMD values are reported as grams per square centimeter (g/cm²). The equipment was calibrated daily with a phantom provided by the manufacturer. BMD was expressed as absolute value as well as *Z*-score and *T*-score. The scores were adjusted to sex and ethnicity. 

### 2.5. Study Endpoints

The primary study endpoints were (1) prevalence of osteoporosis at the first DXA scan performed at our institution and (2) factors influencing changes of BMD during the further disease course. 

The secondary aim was to identify potential risk factors for osteoporosis and its progression during the further disease course, such as steroid use, smoking, BMI changes, operations, therapy changes, vitamin D and calcium supplementation, bisphosphonate therapy or disease activity of CD.

### 2.6. Statistical Analyses

All statistical analyses were performed using IBM SPSS Statistics 24 (Chicago, IL, USA) and SAS 9.4 (Cary, NC, USA). Normality of data distribution was first assessed using the Kolmogorov–Smirnov test. Descriptive statistics were calculated as percentages for discrete variables and presented as medians with ranges—or as averages if the results were normally distributed. To identify potential risk factors of reduced BMD in the index DXA scan, the Mann–Whitney U-test was used for ordinal and continuous variables and x²-tests for categorical variables. Variables with a univariable *p*-value < 0.05 were included in a multivariable analysis. For the multivariable assessment of risk factors for the presence of osteoporosis at first DXA, a logistic regression analysis was performed. To identify potential risk factors influencing the changes in BMD over time, a linear mixed model for repeated measurements was applied with the BMD change as a dependent variable, the previously identified confounders as fixed factors and time of measurement after baseline in years as a random factor. In the linear mixed model for repeated measurements were all patients included with more than one DXA scan. The time periods between single BMD measurements were rounded to full years. BMD measurements performed > 10 years after the index BMD measurement were excluded from the analysis. For the covariance matrix of the model, an autoregressive structure 1st order was used. Parameter estimates were determined together with 95% confidence intervals and associated *p*-values. Due to the exploratory character of the trial, no adjustment for multiple testing was performed, and *p*-values are only to be interpreted descriptively. *P*-values smaller than 0.05 were regarded as statistically significant.

## 3. Results

### 3.1. Patient Characteristics at First BMD Measurement

In total, 576 individual CD patients visited the IBD outpatient clinic between 1 June 2014 and 30 June 2017. Among these, 160 patients underwent only a single BMD measurement, while 180 patients did not undergo any DXA scan mainly due to receiving only a second opinion for the treatment of CD. Three patients were excluded due to minor age. Thus, 393 patients who underwent at least one DXA scan at the IBD outpatient clinic were included in the study. Figure 1 shows a flow diagram of patient inclusion and exclusion. 

Demographics, clinical features and the numbers of DXA scans of the 393 included CD patients are presented in Table 1.

In total, 44.5% of CD patients were males, the median age at diagnosis was 23 years and the median age at the first DXA scan 36 years. The median disease duration of CD until the first DXA scan was 8 years; 37.2% of the patients were smokers; 23.2% of the CD patients had a history of TNFα treatment; 36.4% had an immunomodulator treatment.

### 3.2. Prevalence of Osteopenia and Osteoporosis at First DXA Scan

At their index BMD measurement performed at the outpatient clinic, 157 (39.9%) patients displayed normal BMD, 158 (40.2%) patients were diagnosed with osteopenia and 78 (19.8%) patients suffered from osteoporosis. 

### 3.3. Evaluation of Risk Factors for Osteoporosis

Demographic and clinical parameters were compared between the patients with and without a diagnosis of osteoporosis according to the index DXA scan. Table 2 shows the differences in demographic and clinical data between patients with osteoporosis and those without osteoporosis at the index DXA scan. BMI differed significantly between the groups: The median BMI of patients without osteoporosis was 24.9 kg/m² as compared to 21.3 kg/m² in patients with osteoporosis (*p* < 0.001). Other significant differences were revealed in sex, age at first DXA scan, disease duration at first DXA scan, history of bowel resections, history of anti-integrin treatment and the presence of short bowel syndrome. Male patients suffered from osteoporosis at their first DXA scan more often than female patients (*p* = 0.035). In all, six patients had been diagnosed with short bowel syndrome. The presence of short bowel syndrome was highly associated with osteoporosis (*p* = 0.001) in the univariate analysis.

The occurrence of osteoporosis was not associated with disease extent or age at first diagnosis of CD. However, applying the age categories as defined in the Montreal score, there was a difference concerning the prevalence of osteoporosis (Table 1). Interestingly, in A3 patients according to the Montreal classification, the prevalence of osteoporosis is higher whereas in A1 the rate of osteoporosis is nearly the same. Smoking habits or the presence of extraintestinal manifestations of CD were not associated with the presence of osteoporosis.

Using a multivariable logistic regression model, male sex, older age at first DXA scan, history of bowel resections and reduced BMI were identified as independent risk factors for the presence of osteoporosis at first DXA scan (Table 3).

In Figure 2, the receiver operating characteristics (ROC) curve shows the association of higher BMI in CD patients with higher BMD.

### 3.4. Course of BMD over Time and Relation to CD Therapy

In the cohort of the 233 CD patients who underwent at least two DXA scans, 911 BMD measurements were performed in total, resulting in 588 scan intervals included in the analyses (Figure 1). The numbers of DXA scans performed per patient are presented in Table 1. Demographics, clinical features and the numbers of DXA scans of the 233 included CD patients are presented in the
Appendix A. Comparison of baseline characteristics between the subgroups of patients with osteoporosis versus those without osteoporosis in patients ≥ 2 DXA scans are presented in the Appendix A.

Using a linear mixed model for repeated measurements, the time periods and different patients were analyzed. The results of the linear mixed model for repeated measurements analyzing the BMD change over time are seen in Table 4.

In total, BMD increased in 373 DXA intervals (59.4%), while it remained stable in 19 DXA intervals (3.0%). There was a decrease of BMD in 37.6% of the DXA intervals. In 285 DXA intervals (45.0%), the BMI increased and stayed stable across 194 DXA intervals (30.6%).

Several risk factors were found to be statistically significantly associated with the outcome: male sex, age at DXA scan, presence of an extraintestinal manifestation of CD and higher BMI were associated with an increasing BMD. Age at first diagnosis of CD, disease duration before first DXA scan, first DXA scan result, steroid use and history of immunomodulator treatment were associated with a decreasing BMD. Smoking, calcium or Vitamin D intake, as well as bisphosphonate therapy, were not statistically significantly associated with a change in BMD. In total, there were 41 patients receiving a bisphosphonate therapy and 114 DXA scan time points during which patients were recorded to be under bisphosphonate therapy. For 422 of the DXA scan time points, the intake of calcium preparations was documented, whereas the intake of Vitamin D was documented for 509 DXA scan time points. Active smoking was recorded for 265 time points.

## 4. Discussion

This study is one of the first large retrospective longitudinal studies to explore BMD changes during therapy of CD and osteoporosis, encompassing up to twelve different DXA scans in one patient. As key findings, we found that male sex, history of bowel resections and low BMI are risk factors for low BMD in CD patients. In the longitudinal course, steroid use, history of immunomodulator treatment, female sex and decreased BMI were associated with a decreasing BMD.

At their index BMD measurement, 157 (39.9%) patients displayed normal BMD, 158 (40.2%) patients were diagnosed with osteopenia and 78 (19.8%) patients suffered from osteoporosis. In total, 23.2% of the patients had a history of anti-TNFα treatment and 36.4% a history of immunomodulator treatment prior to the first DXA scan.

In a recent cohort of Polish patients with CD, osteoporosis of the lumbar spine was found in 11.7%, and osteopenia in 36.9% [28]. In an Iranian IBD cohort published in 2006, 5.4% of the patients were diagnosed with osteoporosis, while 26.7% had osteopenia [29].

Data on the prevalence of osteoporosis in Germany is sparse. In the general population over the age of 50, the prevalence of osteoporosis was 8% in males and 33% in females [30].

In comparison, the prevalence of osteopenia and osteoporosis detected in the present study in CD patients appear higher than in the other studies. This result may be explained by the fact that our patients visited a tertiary referral center for the treatment of IBD, which means that they may have presented with more severe illness. Prior to the first DXA scan, 23.2% had a history of anti-TNFα treatment and 36.4% had a history of immunomodulator treatment.

Our study reveals male sex, previous bowel resections and low BMI as risk factors for low BMD in CD patients who visited a tertiary referral center for the treatment of IBD.

Recently, a study from Israel revealed the use of steroids and low BMI as risk factors for low BMD in IBD patients, whereas smoking and male sex reached only borderline significance [12]. In our study, we were able to confirm male sex and low BMI as risk factors for low BMD. Interestingly, smoking was not associated with osteoporosis in our study cohort. In an American cohort of 14,510 people, there was no statistically significant difference in BMD between current smokers and never-smokers in accordance with our data [31].

Evaluating the changes of BMD over time, male sex, the presence of at least one extraintestinal manifestation of CD, higher age at DXA scan and higher BMI were associated with an increasing BMD and thus with a positive disease course.

Patients who gained weight in our study displayed significantly increasing BMD values in the longitudinal course. This underlines the special impact of gaining body weight for patients with CD. There are different potential explanations for why higher BMI is associated with higher BMD. An increasing BMI in CD may be a sign of a decrease in inflammation, which could result in better nutrition and thus in an increasing BMD. Therefore, BMI and BMD changes should be correlated with clinical disease activity. This was not the focus of the present study, but should be investigated further in future projects.

Age at first diagnosis of CD, disease duration to first DXA scan in our outpatient’s clinic, first DXA scan result, steroid use and history of immunomodulator treatment were associated with a decreasing BMD over time. It is common knowledge that the use of steroids results in a reduction of BMD. A meta-analysis found a daily dose of 5 mg of prednisolone or equivalent leading to a reduction in BMD [32]. In this study, we demonstrate that even under osteoporosis therapy with calcium and Vitamin D plus anti-inflammatory IBD therapy, the administration of systemic steroids affects BMD. This stands in contrast to the results from a Spanish retrospective longitudinal study, which revealed that steroid use is not associated with BMD deterioration [33]. Unfortunately, due to the retrospective design of this study, we were not able to elicit the exact doses of steroid treatment which the patients received at the time points of their DXA scans in our study. We recommend careful application even of low doses of steroids over longer treatment periods. Higher initial DXA scan results were also associated with decreasing BMD values in our study. A potential explanation for this finding might be that higher BMD values in the beginning naturally imply the chance of BMD deterioration, while the possibility of an increase is less likely with a higher basic value. Active smoking, calcium or Vitamin D therapy, as well as bisphosphonate therapy, did not significantly influence the course of BMD in CD patients. Our interpretation of this observation is that the positive effects of anti-osteoporotic medications are not sufficient to outweigh the detrimental effects of steroid use in patients with CD.

Efficacy and safety of medical therapies meant to counteract BMD decrease in IBD patients were evaluated in a recent meta-analysis [34]. Contradicting the results of our study, the authors found an increased BMD under bisphosphonate therapy. Similar to our results, the intake of calcium and Vitamin D was shown to be insufficient in preventing BMD decrease in that study [22].

Also in accordance with our results, a recent study showed no improvement in bone health with high dose Vitamin D supplementation; indeed, data showed that it may be harmful [34].

The assumed lack of efficacy of bisphosphonates in our study may be explained by the small number of patients treated with bisphosphonates in our cohort (*n* = 41), which is most likely explained by an avoidance of this class of drugs in young people. In Germany, bisphosphonates are approved only for postmenopausal women or adult men. Therefore, the use of bisphosphonates in younger women or men is off-label.

Unfortunately, the presence of past bone fractures was not sufficiently recorded in our study, so we could not make a reliable statement concerning this matter.

The main strengths of this study are a large number of DXA scans performed over a long period of time, and thus the potential to explore the influence of parallel therapies for CD and osteoporosis in a real-world setting. The data is relatively homogenous as it is restricted to patients in a tertiary center for the treatment of IBD.

Due to the retrospective, uncontrolled character of our study, there is a high risk of confounding. We included only patients visiting a tertiary referral center. In total, 180 patients did not receive a DXA scan due to the fact that their appointment was only for a second opinion, which could result in a selection bias. Further limitations of this study are that patients with lower BMD are overly-represented in this retrospective longitudinal study because patients with initially good BMD values needed no further BMD measurements to be documented in the disease course. Also, we were not able to include serum Vitamin D levels in our study. The time points of first DXA scans in relation to the disease course varied widely between the patients included in the study.

## 5. Conclusions

We conclude from our results that especially in male CD patients, in CD patients with a history of bowel resections and CD patients with low BMI, BMD measurement should be considered in a timely fashion to identify low BMD. If confirmed, these patients have a reasonable chance to increase their BMD by anti-osteoporotic therapy and increase of body weight. Also, we demonstrate that under CD-specific therapy and vitamin D supplementation, steroid use could have the potential to reduce BMD, so this therapeutic option should be applied carefully. Nevertheless, these results have to be verified in further prospective studies.

## Figures and Tables

**Figure 1 jcm-08-02178-f001:**
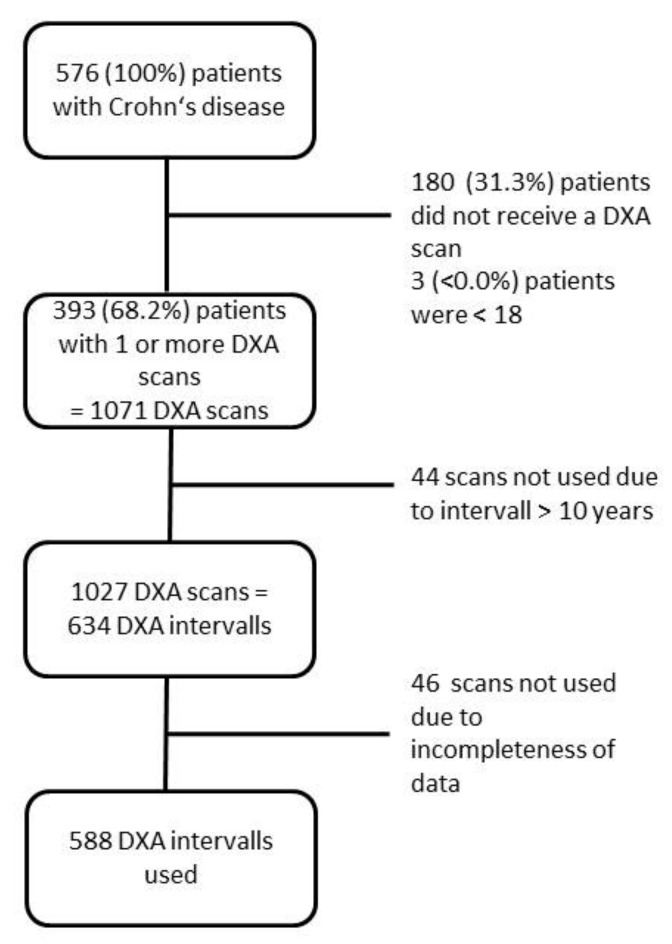
Flow diagram of patient inclusion and exclusion.

**Figure 2 jcm-08-02178-f002:**
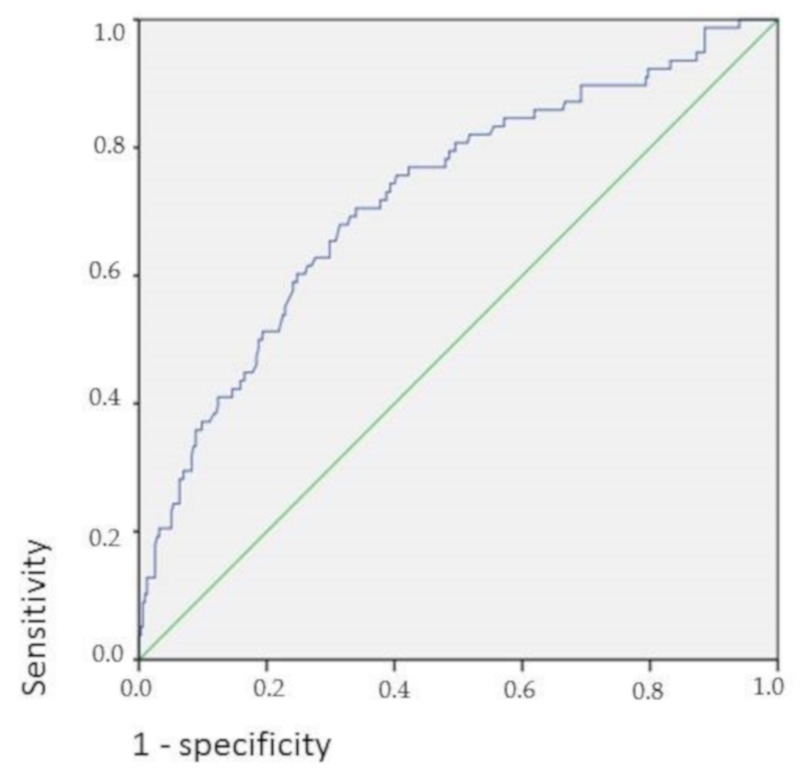
BMI and bone density in Crohn’s disease; Area under the curve: 72.3% (95% CI = 0.658–0.788), *p* < 0.001.

**Table 1 jcm-08-02178-t001:** Baseline characteristics of all included patients with ≥1 dual-energy X-ray absorptiometry (DXA) scan.

Variable	*n* = 393
Male, *n* (%)	175 (44.5)
Age at diagnosis of IBD (years), median (range)	23 (8–68)
Age at first DXA scan (years), median (range)	36 (18–77)
Montreal classification of CD:	
Age, n (A1:A2:A3), *n* = 391	45:302:44
Location, n (L1:L2:L3:L4), *n* = 392	116:57:219:38
Behavior, n (B1:B2:B3), *n* = 392	132:91:168
Disease duration at first DXA scan (years), median (range)	8 (0–61)
Presence of at least one extraintestinal manifestation, *n* (%)	211 (53.7)
Active cigarette smoking at first DXA scan, *n* = 224 (%)	146 (37.2)
BMI (kg/m²), mean ± SD (range)	24.2 ± 5.1 (14.6–45.7)
History of anti-TNFα treatment, *n* (%)	91 (23.2)
History of anti-integrin treatment, *n* (%)	3 (0.8)
History of anti-interleukin treatment, *n* (%)	0 (0)
History of immunomodulator treatment, *n* (%)	143 (36.4)
History of bowel resection(s), *n* (%)	268 (68.2)
Short bowel syndrome, *n* (%)	7 (1.8)
Ostomy, *n* (%)	26 (6.6)
BMD, mean ± SD (range)	0.919 ± 0.136 (0.478–1.362)
BMD lumbar spine, mean ± SD (range)	0.963 ± 0.146 (0.524–1.453)
BMD femur, mean ± SD (range)	0.874 ± 0.153 (0.432–1.430)
BMD according to T-score of WHO	
Normal BMD:osteopenia:osteoporosis (*n*:*n*:*n*)	157:158:78
Number of DXA scans per patient	
1, *n* (%)	160 (40.7)
2, *n* (%)	73 (18.6)
3, *n* (%)	54 (13.7)
4, *n* (%)	39 (9.9)
5, *n* (%)	27 (6.9)
6, *n* (%)	11 (2.8)
7, *n* (%)	8 (2.0)
8, *n* (%)	9 (2.3)
9, *n* (%)	6 (1.5)
10, *n* (%)	2 (0.5)
11, *n* (%)	3 (0.8)
12, *n* (%)	1 (0.3)

BMD: bone mineral density; BMI: body mass index; CD: Crohn’s disease; DXA: dual-energy X-ray absorptiometry; SD: standard deviation; TNFα: tumor necrosis factor alpha; WHO: world health organization; Montreal classification of Crohn’s disease: A1: age < 16 years; A2: age 17–40 years; A3: age > 40 years; L1: location ileal; L2: location colonic; L3: location ileal and colonic; L4: location upper gastrointestinal tract; B1: non-stricturing non penetrating behavior; B2: stricturing behavior; B3: penetrating behavior.

**Table 2 jcm-08-02178-t002:** Comparison of baseline characteristics between the subgroups of patients with osteoporosis versus those without osteoporosis.

	Osteoporosis	No Osteoporosis	*p*-Value
Variable	*n* = 78	*n* = 315	
Male, *n* (%)	43 (55.1)	132 (41.9)	0.035 ^1^
Age at diagnosis of CD (years), median (range)	23 (8–66)	23 (9–68)	0.413 ^2^
Age at first DXA scan (years), median (range)	44 (18–77)	34 (18–75)	0.009 ^2^
Montreal classification of CD:			
Age			0.028 ^1^
A1	10 (13.0)	35 (11.1)	
A2	52 (67.5)	250 (79.6)	
A3	15 (19.5)	29 (9.2)	
Location			0.462 ^1^
L1	24 (31.2)	92 (29.2)	
L2	7 (9.1)	50 (15.9)	
L3	46 (59.7)	172 (54.6)	
L4	5 (6.5)	33 (10.5)	0.290 ^1^
Behavior			0.816 ^1^
B1	25 (32.1)	107 (43.2)	
B2	17 (21.8)	74 (23.6)	
B3	36 (46.2)	132 (42.2)	
Disease duration at first DXA scan (years), median (range)	10.5 (0–61)	8 (0–40)	0.028 ^2^
Presence of at least one extraintestinal manifestation, *n* (%)	42 (53.8)	169 (53.7)	0.975 ^1^
Active cigarette smoking at first DXA scan, *n* (%)	28 (36.8)	118 (38.9)	0.736 ^1^
BMI (kg/m²), mean ± SD (range)	21.3 ± 4.2 (14.6–33.3)	24.9 ± 5.0 (15.2–45.7)	<0.001 ^2^
History of anti-TNFα treatment, *n* (%)	15 (19.2)	76 (24.1)	0.359 ^1^
History of anti-integrin treatment, *n* (%)	2 (2.6)	1 (0.3)	0.042 ^1^
History of immunomodulator treatment, *n* (%)	28 (35.9)	115 (36.5)	0.920 ^1^
History of bowel resection(s), *n* (%)	66 (84.6)	202 (64.1)	0.001 ^1^
Short bowel syndrome, *n* (%)	5 (6.4)	2 (0.6)	0.001 ^1^
Ostomy, *n* (%)	8 (10.3)	18 (5.7)	0.148 ^1^

BMI: body mass index; CD: Crohn’s disease; DXA: dual-energy X-ray absorptiometry; SD: standard deviation; TNFα: Tumor necrosis factor alpha; ^1^ Chi-squared test; ^2^ Mann–Whitney test; Montreal classification of Crohn’s disease: A1: age < 16 years; A2: age 17–40 years; A3: age > 40 years; L1: location ileal; L2: location colonic; L3: location ileal and colonic; L4: location upper gastrointestinal tract; B1: non-stricturing non penetrating behavior; B2: stricturing behavior; B3: penetrating behavior.

**Table 3 jcm-08-02178-t003:** Results of the multivariable logistic regression model to identify risk factors for osteoporosis in Crohn’s disease patients.

Parameter	Odds Ratio (OR)	95% CI	*p*-Value
Male sex	2.511	(1.377; 4.576)	0.003
Age at first DXA scan (years)	1.053	(1.026; 1.081)	<0.001
Disease duration at first DXA scan (years)	0.985	(0.953; 1.018)	0.372
BMI (in kg/m²)	0.761	(0.697; 0.831)	<0.001
History of anti-integrin treatment	2.800	(0.164; 47.729)	0.477
History of bowel resection(s)	3.253	(1.514; 6.989)	0.003
Short bowel syndrome	2.783	(0.371; 20.887)	0.320

BMI: body mass index; DXA: dual-energy X-ray absorptiometry

**Table 4 jcm-08-02178-t004:** Results of the mixed-model analysis to identify factors associated with changes in bone mineral density over time.

Variable	Estimate	95% CI	*p*-Value
Male sex	0.025	(0.014; 0.036)	<0.001
Age at diagnose of CD (in years)	−0.003	(−0.005; −0.001)	0.001
Age at DXA scan (in years)	0.002	(<0.001; 0.004)	0.023
Montreal classification of CD:			
L1	0		
L2	0.006	(−0.012; 0.024)	0.529
L3	−0.001	(−0.013; 0.011)	0.833
B1	0		
B2	−0.012	(−0.027; 0.004)	0.132
B3	−0.011	(−0.027; 0.003)	0.122
Disease duration to first DXA scan	−0.003	(−0.005; −0.001)	0.018
First bone density	−0.180	(−0.119; −0.232)	<0.001
Presence of at least one extraintestinal manifestation	0.013	(0.002; 0.024)	0.002
BMI	0.002	(<0.001; 0.003)	0.011
History of anti-TNFα treatment	−0.012	(−0.026; 0.003)	0.118
History of immunomodulator treatment	−0.016	(−0.028; −0.003)	0.012
History of bowel resections	0.013	(0.000; 0.028)	0.050
Steroid treatment during interval	−0.011	(−0.021; −0.001)	0.028
Smoking during interval	−0.007	(−0.021; 0.006)	0.244
Calcium during interval	0.006	(−0.008; 0.019)	0.401
Vitamin D during interval	−0.009	(−0.023; 0.005)	0.219
Bisphosphonates during interval	−0.011	(−0.026; 0.002)	0.094

BMI: body mass index; CD: Crohn’s disease; DXA: dual-energy X-ray absorptiometry; TNFα: Tumor necrosis factor alpha; Montreal classification of Crohn’s disease: L1: location ileal; L2: location colonic; L3: location ileal and colonic; L4: location upper gastrointestinal tract; B1: non-stricturing non penetrating behavior; B2: stricturing behavior; B3: penetrating behavior.

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
