# Peer review of "Prevalence, Risk Factors and Course of Osteoporosis in Patients with Crohn’s Disease at a Tertiary Referral Center"

_jcm, 2019, doi:10.3390/jcm8122178_

Round 1
Reviewer 1 Report
The authors have performed an observational study on patients with Crohn's disease seen at a tertiary referral centre and used patients with Crohn's disease and at least 2 DXA readings to identify the prevalence of osteoporosis and identify risk factors associated with osteoporosis development. They found a higher prevalence of osteoporosis and osteopaenia than reported at other centres for patients with Crohn's disease and found male sex and low BMI were associated with development of osteoporosis.
The numbers recruited in the study are impressive, but it does suffer substantially due its retrospective and uncontrolled nature which does increase the risk of confounding. The authors should further discuss potential limitations due to the design and avoid overinterpretation of the data in the conclusion.
Major points
Steroid treatment - was oral budesonide included as corticosteroid treatment? Which bone was used for DEXA values? What was done if patients had more than 2 DXA scans performed? Which ones were used for the analysis? Diagnosis of Crohn's disease - there should be mention of how the diagnosis was arrived at What was the median follow up time in the cohort? What was the median time between DEXAs? Was this done at a set time or at the discretion of clinicians 180 patients did not undergo DEXA. Are the reasons for this known? Do you think this causes selection bias or may impact on prevalence estimates? This should be added in the discussion, Results 3.2For the prevalence estimate of osteoporosis any patient who had one DXA could be included. Why were people with 2 DXA only included and was there a difference in results between those who had 2 or 1 study performed? Results 3.3
Is this for participants with an index DXA showing osteoporosis or people who develop osteoporosis during the study?
Minor points
Line 55 "T" should be "T score"
Line 57/58 replace "- or patients under corticosteroid treatment -" with ", or patients treated with long-term corticosteroids,"
Line 76 - consider changing monocentric to single centre
Line 76 last word should be a not the
Line 81 first word at should be to
Line 85 "daily patient visiting lists" is difficult to understand, consider revising
Line 100 consider changing under to on
Line 145 avoid use of "-"
Section 3.1 - please add both % and numbers for figures given
Table 2 - with the Montreal classification please also add the definition
Line 103 please elaborate on the further risk factors for which people were considered for bisphosphonates
Line 191 first word should be "a" not "the"
Line 262 - it should be "before the first DXA..."
Line 275-276 - you should elaborate on the impact of gaining BMI. For example, do you think it reflects better nutrition or less inflammation?
Line 277 - first DXA scan needs to be further explained or made a dichotomous variable
Reviewer 2 Report
Abstract:
Length was reasonable and the background and aims were clearly stated.Introduction:
I would recommend more up-to-date references, such as the association between low BMD and IBD, they could reference in newer study Dig Dis 2019; 37: pp. 284-290. Minor edits such as: in line 39- remove "by" from "by 10%", line 57- change "under" to "receiving" , line 59- remove "that" from "that it is of great", line 60- remove "mostly", and line 70- change "prophylaxis" to "prevention". Line 47: "Furthermore, genetic factors are discussed", this sentence needs to be elaborated further.Methods:
With the study design, I am concerned that excluding patients that had less than 2 BMDs, will skew the prevalence of osteopenia and osteoporosis in the study and will only include patients who are with osteopenia or osteoporosis, because technically these patients will be receiving another DEXA scan within the next 10 years. Therefore this will over inflate the prevalence of osteopenia and osteoporosis in this cohort. I think they should include all the BMDs done for the purpose of their primary endpoint of the study (prevalence of low BMD in CD and risk factors of low BMD). For the purposes of predictors of changes in BMD over time, in that case they would only include patients who have had more than 2 BMDs. I believe it is very important to include the dose of steroids that these patients have received and the duration of steroid treatment. Typically patients who have received less than 3 months of steroids, per current ACG and AGA guidelines, do not need screening for osteoporosis. There for duration and dose can play a role in change in BMD. The authors acknowledged the importance of dose and duration of steroid use in low BMD in their discussion, but this does not translate into their methods. It is not clear why the parameters with p<0.1 were included in the logistic regression if the cut off for statistical significance was set at p<0.05.Results:
Although the authors state that they included patients Age >/= 18, their results (table 2 show that the age at first DEXA scan range of their cohort go from 13-77 in the osteoporosis group and 15-67 in the no osteoporosis group indicating that there were patients <18 included. Table 3: OR for Male Sex is 0.290, which suggests male sex is protective for osteoporosis in CD patients (contrary to what is stated in the text). Similarly for short bowel syndrome OR is 0.262. Consider revising the analysis.Line 301 change "adversion" to "aversion"/ Line 304 remove "well"
Round 2
Reviewer 1 Report
The authors have made substantial changes to the manuscript and addressed the suggested changes. There are still a number of spelling and grammatical errors that need to be corrected.
Line 50 consider changing “hints” to “suggestions” and remove the comma after the word
Line 60 and 61 a comma (,) is more formal than an em dash (-) and they should be replaced in this sentence
Line 155 remove “smaller than”
Line 188 change “structuring” to “stricturing”
Table 2 and Table 4 – include the definition of each of the Montreal Classification variables with names in the table to help readers understand this
Line 304 change the spelling of “their”
Line 321 consider change “aversion to” to “avoidance of”
Author Response
Response to Reviewer 1 Comments
Comments and Suggestions for Authors
The authors have made substantial changes to the manuscript and addressed the suggested changes. There are still a number of spelling and grammatical errors that need to be corrected.
Line 50 consider changing “hints” to “suggestions” and remove the comma after the word
Line 60 and 61 a comma (,) is more formal than an em dash (-) and they should be replaced in this sentence
Line 155 remove “smaller than”
Line 188 change “structuring” to “stricturing”
Table 2 and Table 4 – include the definition of each of the Montreal Classification variables with names in the table to help readers understand this
Line 304 change the spelling of “their”
Line 321 consider change “aversion to” to “avoidance of”
Response: All minor points have been changed in the text of the new version of the manuscript.
Reviewer 2 Report
Thank you for all the responses and changes made to the manuscript.
Author Response
Response to Reviewer 2 Comments
Comments and Suggestions for Authors
Thank you for all the responses and changes made to the manuscript.
Response: Thank you for your comments and suggestions.